# BIGSAGE: UNSUPERVISED INDUCTIVE REPRESENTATION LEARNING OF GRAPH VIA BI-ATTENDED SAMPLING AND GLOBAL-BIASED AGGREGATING

## ABSTRACT

Different kinds of representation learning techniques on graph have shown significant effect in downstream machine learning tasks. Recently, in order to inductively learn representations for graph structures that is unobservable during training, a general framework with sampling and aggregating (GraphSAGE) was proposed by Hamilton and Ying and had been proved more efficient than transductive methods on fileds like transfer learning or evolving dataset. However, GraphSAGE is uncapable of selective neighbor sampling and lack of memory of known nodes that've been trained. To address these problems, we present an unsupervised method that samples neighborhood information attended by co-occurring structures and optimizes a trainable global bias as a representation expectation for each node in the given graph. Experiments show that our approach outperforms the state-of-the-art inductive and unsupervised methods for representation learning on graphs.

## 1 INTRODUCTION

Graphs and networks, e.g., social network analysis Hamilton et al. (2017a), molecule screening Duvenaud et al. (2015), knowledge base reasoning Trivedi et al. (2017), and biological protein-protein networks (Zitnik & Leskovec (2017)), emerge in many real-world applications. Learning low-dimensional vector embeddings of nodes in large graphs has been proved effective for a wide variety of prediction and graph analysis tasks (Grover & Leskovec (2016); Tang et al. (2015)). The high-level idea of node embedding is to explore high-dimensional information about the neighborhood of a node with a dense vector embedding, which can be fed to off-the-shelf machine learning approaches to tasks such as node classification and link prediction (Perozzi et al. (2014)).

Whereas previous approaches (Perozzi et al. (2014); Grover & Leskovec (2016); Tang et al. (2015)) can transductively learn embeddings on graphs, without re-training they cannot generalize to new nodes that are newly added to graphs. It is ubiquitous in real-world evolving networks, e.g., new users joining in a social friendship circle such as facebook. To address the problem, Hamilton et al. (2017b) propose an approach, namely GraphSAGE, to leverage node feature information (e.g., text attributes) to efficiently generate node embeddings for previously unseen nodes. Despite the success of GraphSAGE, it randomly and uniformly samples neighbors of nodes, which suggests it is difficult to explore the most useful neighbor nodes. It could be helpful if we can take advantage of the most relevant neighbors and ignore irrelevant neighbors of the target node. Besides, GraphSAGE only focuses on training parameters of the hierarchical aggregator functions, but lose sight of preserving the memory of the training nodes, which means when training is finished, those nodes that have been trained over and over again would still be treated like unseen nodes, which causes a huge waste.

To address *the first issue*, inspired by GAT (Velickovic et al. (2017)), a supervised approach that assigns different weights to all neighbors of each node in each aggregating layer, we introduce a bi-attention architecture (Seo et al. (2016)) to perform selective neighbor sampling in unsupervised learning scenarios. In unsupervised representation learning, when encoding embeddings of a positive[1] node pair before calculating their proximity loss (Hamilton et al. (2017a)), we assume that

---

[1]In random-walk based approaches, nodes that tend to cooccur in short random walks over the graph are usually referred as positive to each other.

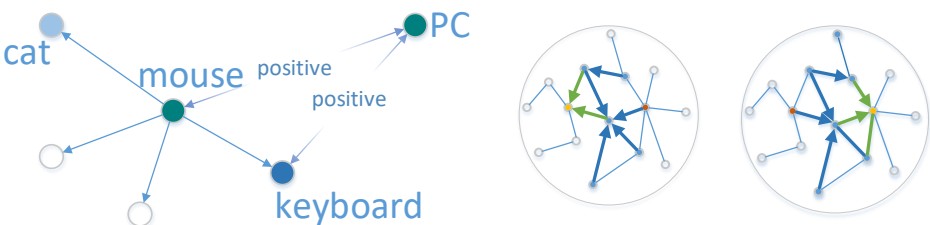

Figure 1: Visual illustration of the co-occurrence bi-attented sampling of nodes' neighborhood. The left one figure gives a vivid example of why we can use co-occurrence node to filter out useful information; and the two figures in the right show how attention from co-occurrence nodes affects sampling and aggregating over the neighborhood.

neighbor nodes positive to both of the pair should have larger chance to be selected, since they are statistically more relevant to the current positive pair than other neighbors. For example, when embedding words like "mouse", in Figure 1, it's more reasonable to choose "keyboard" rather than "cat" as sampled neighbor while maximizing co-occrrence probability between "mouse" and "PC", because "keyboard" also tends to co-occurr with "PC", which means its imformation should be more relevant. We thus stack a bi-attention architecture (Seo et al. (2016)) on representations aggregated from both side in a positive node pair. In this way, we learn the most relevant representations for each positive node pair corresponding to their most relevant neighbors, and simply use a fixed-size uniform sampling which allows us to efficiently generate node embeddings in batches.

To address *the second issue*, we combine the idea behind transductive approaches and inductive approaches, by intuitively applying an additive global embedding bias to each node's aggregated embedding. The global embedding biases are trainable as well as parameters of aggregator functions and can be considered as a memorable global identification of each node in training sets. When the training is completed, we generate the embedding for each node by calculating an average of multiple embedding outputs corresponding to different sampled neighbors with respect to different positive nodes. In this way, nodes that tend to co-occur in short random-walks will have more similar embeddings based on our bi-attention mechanism.

Based on the above-mentioned two techniques, we propose a novel approach, called BIGSAGE (which stands for the **BI**-attention architeture, global **BI**as and the original framework **G**raph**SAGE**,) to explore most relevant neighbors and preserve previously learnt knowledge of nodes by utilizing bi-attention architecture and introducing global bias, respectively.

## 2 RELATED WORK

### 2.1 NETWORK EMBEDDING

In this paper, we focus on unsupervised and inductive node embedding learning for large and evolving network data. For unsupervised learning, many different approachs have been proposed. Deep-Walk (Perozzi et al. (2014)) and node2vec (Grover & Leskovec (2016))are two classic approaches that learn node embeddings based on random-walks using or extending the Skip-Gram model. Similarly, LINE (Tang et al. (2015))seeks to preserve first- and second-order proximity and trains the embedding via negative smpling. SDNE(Wang et al. (2016)) jointly uses unsupervised components to preserve second-order proximity and expolit first-order proximity in its supervised components.

Unlike the methods mentioned above, some approaches were proposed to takes use of not only network structure but also node attributes and potentially node labels. Such as TRIDNR(Pan et al. (2016)), CENE(Sun et al. (2016)), TADW(Yang et al. (2015)). GraphSAGE(Hamilton et al. (2017b)), which this paper is motivated from, also requires rich attributes of nodes for sampling and aggregating into embeddings that preserve rich local neighborhood strutural infromation.

Recently, in order to address the problem in large and dense networks of consequently encounted newly jointed nodes/edges, approaches such as Hamilton et al. (2017b) Velickovic et al. (2017)

Bojchevski & Günnemann (2017) were proposed as inductive ways of graph embedding learning and had produced impressive performance across several large-scale inductive benchmarks. In fact, we find that the key of inductive learning is to learn an embedding encoder that relies on only information from a single node itself and/or its local observable neighborhood, instead of the entire graph as in transductive setting.

## 2.2 ATTENTION

Attention mechanism in neural processes have been largely studied in Neuroscience and Computational Neuroscience (Itti et al. (1998); Desimone & Duncan (1995)) and since these few years frequently applied in Deep Learning for speech recognition (Chorowski et al. (2015)), translation(Luong et al. (2015)), question answering (Seo et al. (2016)) and visual identification of objects Xu et al. (2015) . The principle inside attention mechanism is that focusing on most pertinent parts of the input, rather than using all available information, a large part of which being irrelevant to compute the desirable output. In this paper, we are inspired by Seo et al. (2016) to construct a bi-attention layer upon aggregators in order to capture the useful part of the neighborhood.

## 3 MODEL

In this section, we propose a hierarchical bi-attended sampling and global-biased aggregating framework (BIGSAGE). We start by presenting an overview of our framework: the training in Algorithm 1 and the embedding generation in Algorithm 2. Followingly, section 3.2 gives the detailed implemention of our bi-attention architeture, section 3.3 demonstrates how we combine global bias within the framework. Given an undirected network $\mathcal{G} = \{V, E, \boldsymbol{X}\}$, in which a set of nodes $V$ are connected by a set of edges $E$, and $\boldsymbol{X} \in \mathbb{R}^{|V| \times f}$ is the attribute matrix of nodes. We denote the global embedding bias matrix as $\boldsymbol{B} \in R^{|V| \times d}$, where each row of $\boldsymbol{B}$ represents the d-dimensional global embedding bias of each node. The hierarchical layer number is set as $K$, the embedding output of $k$-th layer is represented by $\boldsymbol{h}^k$, and the final output embedding $\boldsymbol{z}$.

## 3.1 OVERVIEW

To learn representations in unsupervised setting, we apply the same graph-based loss function used in the origin GraphSAGE:

$$J_{\mathcal{G}}(z_v) = -log(\sigma(\boldsymbol{z}_v^T \boldsymbol{z}_{v_p})) - Q \cdot \mathbb{E}_{v_n \sim P_n(v)} log(\sigma(-\boldsymbol{z}_v^T \boldsymbol{z}_{v_n}))$$

where node $v_p$ co-occurs with $v$ on fixed-length random walk (Perozzi et al. (2014)), $sigma$ is the sigmoid function, $P_n$ is a negative sampling distribution, and Q defines the number of negative samples.

Algorithm 1 shows the training of our framework.

---

**Algorithm 1** BIGSAGE: training

**input:** Training graph $\mathcal{G}_{train}(V_{train}, E_{train})$; node attributes $\boldsymbol{X}$; global embedding bias matrix $\boldsymbol{B}$; sampling times $T$

1: $zero\_initialize(\boldsymbol{B})$
2: $\boldsymbol{h}_v^0 \leftarrow \boldsymbol{x}_v, \forall v \in V_{train}$
3: run random-walk in $\mathcal{G}_{train}$ and do negative sampling to gain a set of triplets: $\{(v, v_p, v_n)\}$
4: **for** $(v, v_p, v_n)$ **do**
5: $\quad \mathbb{Z}_1 \leftarrow \emptyset, \mathbb{Z}_2 \leftarrow \emptyset$
6: $\quad$ **for** $t \in \{1, ..., T\}$ **do**
7: $\quad\quad \mathbb{Z}_1 \leftarrow \mathbb{Z}_1 \cup \{\text{SAGB(v)}\}$
8: $\quad\quad \mathbb{Z}_2 \leftarrow \mathbb{Z}_2 \cup \{\text{SAGB(v}_\text{p})\}$
9: $\quad \boldsymbol{z}_v, \boldsymbol{z}_{v_p} \leftarrow \text{BIATT}(\mathbb{Z}_1, \mathbb{Z}_2)$
10: $\quad \boldsymbol{z}_{v_n} \leftarrow \text{SAGB(v}_\text{n})$
11: $\quad$ calculate the graph-based loss $J_{\mathcal{G}}$ and update model parameters with SGD

---

When generating embeddings with the learned parameters after optimization is done, GraphSAGE encode only one single random-partial-sampled neighborhoods for each node, likely leaving out information of those unselected part.

To fully preseve the structural information around, we first rerun random walk on a full graph that inlcudes the seen and unseen nodes; Then, through our bi-attention mechanism built upon the aggregating layers, we generate the most relevant embeddings of each node w.r.t its positive nodes; Eventually, we take average of these generated embeddings as final embeddings of each node and use them in downstream machine learning tasks. The generation process is shown in Algorithm 2:

---

**Algorithm 2** BIGSAGE: generation of embedding

---

**input:** Testing graph $\mathcal{G}_{test}(V_{test}, E_{test})$; node attributes $\boldsymbol{X}_{test}$; learned model BIGSAGE
**output:** Vector representations $\boldsymbol{z}_v$ for all $v \in V_{test}$

1: run random-walks for each node in $\mathcal{G}_{test}$ and gain a set of positive nodes pair:$\{(v, v_p)\}$
2: $\mathbb{Z}_i \leftarrow \emptyset, \forall i \in V_{test}$
3: **for** $i, j \in \{(v, v_p)\}$ **do**
4:      use step 5-9 in Algorithm 1 to generate $\boldsymbol{z}_i, \boldsymbol{z}_j$
5:      $\mathbb{Z}_i \leftarrow \mathbb{Z}_i \cup \{\boldsymbol{z}_i\}$
6:      $\mathbb{Z}_j \leftarrow \mathbb{Z}_j \cup \{\boldsymbol{z}_j\}$
    **return** $\boldsymbol{z}_v \leftarrow \text{MEAN}(\mathbb{Z}_v), \forall v \in V_{test}$

---

## 3.2 CO-OCCURRENCE BI-ATTENDED "SAMPLING"

We now describe our bi-attention achitecture. Given node $n$, and node $m$ as a positive node pair, after sampling $T$ times using a uniform fixed-size sampler, we have

$$\boldsymbol{h}_{nt}^K = A_{gg}(\boldsymbol{h}_n^{K-1}, \{\boldsymbol{h}_{j \in N^t(n)}^{K-1}\})$$

$$\boldsymbol{h}_{mt}^K = A_{gg}(\boldsymbol{h}_m^{K-1}, \{\boldsymbol{h}_{j \in N^t(m)}^{K-1}\})$$

$$t = 1, ..., T$$

With $T$ different representations corresponding to $T$ different sampled neighborhoods, their similarity matrix can be calculated by

$$S_{i,j} = \alpha(\boldsymbol{h}_{ni}^K, \boldsymbol{h}_{mj}^K), i, j = 1, ..., T$$

where $\alpha$ represents an dot-product operation.

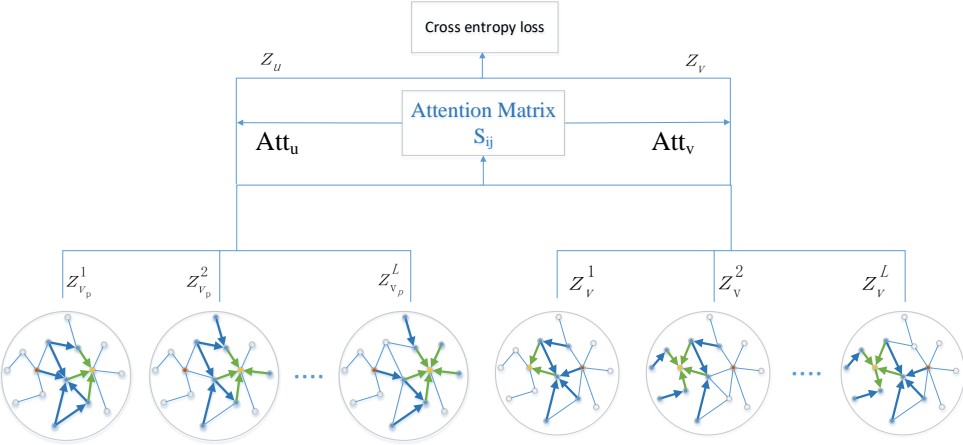

Figure 2: Bi-attention layer between the final aggregating layer and loss layer.

our goal is to find the most similar or relevant neighborhood match between n and m within $T \times T$ possibilities, so we need to apply softmax on the flattened similarity matrix, and sum up by collumn(row) to gain attention over $T$ neighborhoods of n(m):

$$\boldsymbol{att}_n = reduce\_sum(softmax(\boldsymbol{S}), 0),$$

$$\boldsymbol{att}_m = reduce\_sum(softmax(\boldsymbol{S}), 1),$$

and apply attention to $T$ Kth layer representations for the final encoded embeddings,

$$\boldsymbol{h}_n = \sum_{t=1}^{T} \boldsymbol{att}_{nt} \boldsymbol{h}_{nt}^K$$

$$\boldsymbol{h}_m = \sum_{t=1}^{T} \boldsymbol{att}_{mt} \boldsymbol{h}_{mt}^K$$

The aggregation process with bi-attention architecture is illustrated by Figure 2.

## 3.3 GLOBAL BIAS

At first, we consider simply adding a trainable bias upon the encoder's final ouput for each node in training set,

$$\boldsymbol{z}_i = \boldsymbol{h}_i^K + \boldsymbol{b}_i$$

$$\boldsymbol{b}_i = one\_hot(i)^T \boldsymbol{B}$$

By training global bias, Our framework will be able to learn parameters of aggregator functions for inductive learning meanwhile preserve sufficient embedding informations for known nodes. On one hand, these informations can be reused to produce embeddings for the known nodes or the unknown connected with the known, as supplement to the aggregator. On the other hand, they can patially offset the uncertainty of the generation braught by the random sampling. But through further research, we find it more efficient when appling global bias to not only the last layer but also all hidden layers, as follows:

$$\boldsymbol{h}_i^{k+1} = \boldsymbol{W}^{k+1}(\boldsymbol{h}_i^k, \{\boldsymbol{h}_{j\in N(i)}^k\}) + \boldsymbol{b}_{agg}^{k+1} + \boldsymbol{b}_i$$

$$\boldsymbol{b}_i = one\_hot(i)^T \boldsymbol{B}$$

$$\boldsymbol{z}_i \leftarrow \boldsymbol{h}_i^K$$

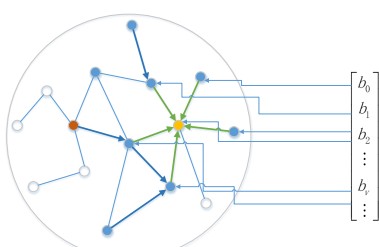

Figure 3: Applying global bias through aggregating .

Applying globla bias vectors through all layers improves not only the expressivity of representations on hidden layers, but also the training efficiency. In fact, the hidden layers' output embeddings and the global biases are now belonging to one single d-dimensional vector space, as a result of which, the parameters from lower layers can be updated more directly by the involving global bias vectors that are adjusted for all different layers including the last layer, which means the loss funtion can now have more instant impact on the lower layers instead of back-propagating from top to bottom.

We propose our aggregating process with global bias in Algorithm 3.

---

**Algorithm 3** SAGB: sampling and aggregating with globa_bias

---

**input:** node $u$; hierarchical depth $K$; weight matrices $\boldsymbol{W}^k$; non-linearity $\sigma$; differentiable neighbor aggregator $\mathrm{A}_{gg}^k$; fixed-size uniform sampler $S : v \to 2^V$
**output:** embedding $\boldsymbol{z}_u$;

1: $\mathbb{N}_u^0 = \{u\}$;
2: **for** $l = 1...K$ **do**
3:     $\mathbb{N}_u^l \leftarrow \{S(i), \forall i \in \mathbb{N}_u^{l-1}\}$;
4: $\mathbb{N}_u = \mathbb{N}_u^0 \bigcup \mathbb{N}_u^1 \bigcup ... \bigcup \mathbb{N}_u^K$;
5: **for** k=1...K **do**
6:     **for** $j \in \mathbb{N}_u^0 \bigcup \mathbb{N}_u^1 \bigcup ... \bigcup \mathbb{N}_u^{K-k}$ **do**
7:         $\boldsymbol{h}_j^k \leftarrow \sigma(A_{gg}^k(\boldsymbol{h}_j^{k-1}, \{\boldsymbol{h}_m^{k-1}, \forall m \in S^*(j)\}, \boldsymbol{W}^k)) + one\_hot(j)^T \boldsymbol{B}$
    **return** $\boldsymbol{z}_u \leftarrow \boldsymbol{h}_u^K$

---

# 4 EXPERIMENTS

In this section, we evaluate BIGSAGE against strong baselines on three challenging benchmark tasks: (i) classifying Reddit posts as belonging to different community; (ii) predicting different classes of papers in Pubmed (Sen et al. (2008)); (iii) classifying protein functions across varieties of protein-protein interaction (PPI) graphs (Zitnik & Leskovec (2017)). We start by summarizing the overall settings in our comparison experiments, and then present the experiment results of each task. We also study the seperate effect of our bi-attention architecture and global bias in section 4.4.

## 4.1 EXPERIMENTAL SETTINGS

We compare BIGSAGE against the following approaches for graph representations learning under a fully unsupervised and inductive setting:

- GraphSAGE: Our proposed method originates from the unsupervised variant of Graph-SAGE, a hierarchical sampling and aggregating framwork for inductive learning. We compare our method against GraphSAGE using three different aggregator: (1) Mean aggregator, which simply takes the elementwise mean of the vectors in $h_{u \in N(v)}^{k-1}$; (2) LSTM aggregator, which adapts LSTMs to encode a random permutation of a node's neighbors' $h^{k-1}$; (3) Maxpool aggregator, which apply an elementwise maxpooling operation to aggregate information across the neighbor nodes.

- Graph2Gauss(Bojchevski & Günnemann (2017)): Unlike GraphSAGE and my method, G2G only uses the attributes of nodes to learn their representations, with no need for link information. Here we compare against G2G to prove that certain trade-off between sampling granularity control and embedding effectiveness does exists in inductive learning scenario.

- SPINE, Guo et al. (2018): Instead of hierarchical neighbor sampling, SPINE uses Rooted-PageRankLiben-Nowell & Kleinberg (2007) to represent the high-order structural proximities of neighborhood. The k largest proximities are then employed to aggregate attributes of the corresponding k neighbor nodes.

For our proposed approach and the origin framework, we set the number of hierarchical layers as $K = 2$ with neighbor sampling sizes $S_1 = 20$ and $S_2 = 10$, the number of random-walks for each node as 100 and the walk length as 5. The sampling time of our bi-attention layer is set as $T = 10$. For all methods, the dimensionality of embeddings is set to 256. Our approach is impemented in Tensorflow (Abadi et al. (2016)) and trained with the Adam optimizer (Kingma & Ba (2014)) at an initial learning rate of 0.0001. We report the comparison results in Table 1.

## 4.2 INDUCTIVE LEARNING IN EVOLVING GRAPHS

In real-world large and evolving graphs, inductive node embedding learning techniques would require high efficiency of the information extraction strategy as well as stability and robustness of the

Table 1: Prediction results for the three datasets (micro-averaged F1 scores).

|  | Reddit | Pubmed | PPI |
|---|---|---|---|
| Graph2Gauss | - | 0.789 | 0.430 |
| SPINE | - | - | 0.505 |
| GraphSAGE-mean | 0.897 | 0.820 | 0.502 |
| BIGSAGE-mean | **0.927** | **0.834** | **0.573** |
| GraphSAGE-seq | 0.907 | 0.811 | 0.505 |
| BIGSAGE-seq | 0.903 | **0.828** | **0.563** |
| GraphSAGE-pool | 0.892 | 0.830 | 0.510 |
| BIGSAGE-pool | **0.915** | **0.843** | **0.566** |

embedding encoder. We here perform comparison against other approaches to show our method's effective use in these challenging datasets.

Reddit is an large internet forum where users can post and comment on any content they are interested in. The task is to predict the community, that a post belongs to. We use the exact dataset conducted by Hamilton et al. (2017b). In this dataset, each node represents a post and are connected with one another if the same user comments on both of them. The node attribute is constructed by word2vec embeddings of post contents. The first 20 days is for training, and the rest for testing/validation. In all, this dataset contains 232,965 nodes(posts) with an average degree of 492. The first collumn summarizes the comparison results against GraphSAGE. For LSTM aggregator, our model shows slightly poorer performance, which is reasonable, because LSTM aggregator causes more differences between multiple sampled neighborhoods, not only the components but also the order, making it harder for our bi-attention layer to capture the proximities between neighborhoods. One can observe that BIGSAGE outperforms GraphSAGE in both mean and pool aggregator.

Another representative of evolving graphs we evaluate on is Pubmed, one of the commonly used citation network data . This dataset contains 19717 nodes and 44324 edges. We remove 20 percengt of the nodes as unseen, the rest for training. From the second column of Table 2 we find our model have better prediction results in all three aggregators.

### 4.3 GENERALIZING ACROSS GRAPHS

Generalizing accross graphs requires inductive methods capable of learning a transferable encoder function rather than the present community structure.

The protein-protein-interaction(PPI) networks dataset consists of 24 graphs corresponding to different human tissues(Zitnik & Leskovec (2017)). We use the preprocessed data provided by Hamilton et al. (2017b), where 20 graphs for training, 2 for validation and 2 for testing. For each node, there are 50 features representing their positional gene sets, motif gene sets and immunological signatures, and 121 labels set from gene ontology( collected from the Molecular Signatures Database (**?**)). This dataset contains 56944 nodes and 818716 edges.

The final collumn of Table 1 shows us that our method outperforms GraphSAGE by 14% at most on the PPI data. The results on three different aggregators indicates that the Mean-aggregator beats the other two in our method. And we also quote the micro-averaged F1 score of SPINE which is based on the exact same PPI dataset as ours.

### 4.4 MODEL STUDY

In this section we adjust BIGSAGE for tests on PPI to further study the seperate effect of bi-attention layer and global bias:

- BIGSAGE-ba: only with bi-attention layer, no global bias;
- BIGSAGE-sg: with bi-attention layer, global bias only applied to embeddings of the last layer;
- BIGSAGE-cb: with bi-attention layer, global bias applied to all layer during training but forgotten (reset to zero matrix) while generating embeddings.

Table 2: Model Study

|            | PPI       |
|------------|-----------|
| GraphSAGE  | 0.502     |
| BIGSAGE-ba | 0.521     |
| BIGSAGE-sg | 0.518     |
| BIGSAGE-cb | 0.523     |
| BIGSAGE    | **0.572** |

From Table 2, we observe the three variant of BIGSAGE still show certain advance to the origin GraphSAGE, but reasonably less accurate than the origin BIGSAGE. The result of using only bi-attention layer proves the effect of bi-attended sampling. And the comparison between BIGSAGE-sg and BIGSAGE shows the high efficiency of applying global bias through all layers. From the result of BIGSAGE-cb, we find that even after training with global-bias, it's critical to have the memory of the known nodes, which is stored in the learnt global embedding biases.

## 4.5 TRADE-OFF

We compare Graph2Gauss against our model as well as GraphSAGE on Pubmed and PPI. Note that Graph2Gauss only needs node attributes for embedding learning. The comparison results in Table 1 shows that our model and GraphSAGE beat g2g whether in evolving network data or generalizing across different graphs, which proves the significance of neighbor sampling for inductive learning and that certain trade-off exists between encoding granularity and embedding effect.

## 5 CONCLUSIONS

In this paper, we proposed BIGSAGE, an unsupervised and inductive network embedding approach which is able to preserve local proximity wisely as well as learn and memorize global identities for seen nodes while generalizing to unseen nodes or networks. We apply a bi-attention architeture upon hierarchical aggregating layers to directly capture the most relevant representations of co-occurring nodes. We also present an efficient way of combining inductive and transductive approaches by allowing trainable global embedding bias to be retrieved in all layers within the hierarchical aggregating framework. Experiments demenstrate the superiority of BIGSAGE over the state-of-art baselines on unsupervised and inductive tasks.

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
