# OpenReview forum: "BIGSAGE: unsupervised inductive representation learning of graph via bi-attended sampling and global-biased aggregating"
_ICLR.cc/2019/Conference_

### Official Review · AnonReviewer2 · 2018-10-26
**Novelty of the paper seems to be marginal**

**Rating:** 4
**Confidence:** 4

**Review:**

This paper studied learning unsupervised inductive node embeddings with an attention mechanism. For each positive edge, multiple different sets of neighborhoods are sampled for both the source and target nodes, and the similarity between the neighborhood are used as the attention functions. Experimental results prove the effectiveness of the proposed approach over GraphSAGE on a few networks.

Strength:
- learning unsupervised inductive node embeddings is an important problem
- the proposed method seems to work

Weakness:
- the novelty of the proposed method seems to be very marginal
- the experiments are quite weak
- the complexity of the algorithm seems to be very high

Details:
- the complexity of the algorithm seems to be very high seem for each pair of nodes, multiple sets of neighborhoods must be sampled for each node.
- there are also other approaches for inductive unsupervised node embeddings, for example, the varitional graph autoencoder method (Kipf et al. 2017).
- I am wondering how the proposed method performs comparing with the methods of only selecting the nodes which form triangles with the given positive edges.

---

### Official Review · AnonReviewer3 · 2018-10-31
**Overall quality is not high.**

**Rating:** 4
**Confidence:** 3

**Review:**

This paper proposes a new representation learning method for graphs.

Quality:
The quality of the paper is not high due to vague presentation of the proposed method (see clarity).
Moreover, there is no theoretical analysis and empirical evaluation is not thorough (see significance).

Clarity:
This paper is not clearly written and many parts are unclear.
- In Introduction, what are "the first issue" and "the second issue"?
- There are many grammatical mistakes (such as missing articles and the third-person singular -s) and mistakes of mathematical notations.
- Too many symbols are not mathematically defined and it is hard to understand the paper. The current version is not appropriate for publication.

Originality:
The proposed method is a minor extension of the existing method GraphSAGE. Hence the originality is not high.

Significance:
- There is no theoretical analysis of the proposed method. Hence the significance is not high.
  In particular, the advantage of the proposed method compared to the existing approach (GraphSAGE) should be theoretically analyzed.
- How to set parameters in practice? The performance of the proposed method will be greatly affected by parameter setting.
  In experiments, the sensitivity of the proposed method with respect to parameter changes should be analyzed.

Pros:
- The relevant problem is studied.
Cons:
- Presentation is not good.
- Theoretical analysis is not given.
- Empirical analysis is not thorough.

Other comments:
- P.3, L.5 in Sec.3: "G = {V, E, X}" should be "G = (V, E, X)"

---

### Official Review · AnonReviewer4 · 2018-11-07
**An increment of GraphSAGE with restricted applications**

**Rating:** 2
**Confidence:** 4

**Review:**

This paper modifies the GraphSAGE on unsupervised inductive node embedding.
The authors propose to use the bi-attention architecture to sample
interesting nodes (instead of the uniform sampler in GraphSAGE), and to use a
global embedding bias matrix in the local aggregating functions. The method
showed improvements over GraphSAGE and other baselines on unsupervised
graph embeddings.

The proposition makes sense and the performance improvements are expected.

A major comment, however, is that that the proposed method is useful in very
restricted settings, and it is not clear how to generalize to
other applications which GraphSAGE can be applied on.
The overall technical contribution is incremental and
may not have enough novelty to be published in ICLR.

The technical representation is very poor, unorganized and not self-contained.
The paper cannot pass the threshold merely based on the way it is presented.

In the algorithms, please give the output besides the input. After the
algorithms, please remark on the computational and memory cost.

In algorithm 1, what is this function BIATT()?
After algorithm 1, please describe this function as well as SAGE().

In the beginning of section 3, please describe the meaning of the
global bias matrix. In algorithm 1, if B is zero-initialized, why
does one need it as input?

Some of the equations are poor formatted (e.g. reduce_sum in page 5).
Please try to use rigorous mathematical formulations instead of "pseudo equations".
For example, re-write "One_hot(i)". In section 3.2, explain A_{gg}, etc.
use $\langle \rangle$ instead of $\alpha$.

There are many typos.

---

### Meta-Review · Area_Chair1 · 2018-12-15
**Novelty, complexity and poor presentation are all of concern.**

**Confidence:** 5
**Recommendation:** Reject

**Metareview:**

AR2 is concerned about the marginal novelty, weak experiments and very high complexity of the algorithm. AR3 is concerned about lack of theoretical analysis and parameter setting. AR4 is concerned that the proposed method is useful in very
restricted settings and the paper is incremental.

Unfortunately, with strong critique from reviewers regarding the novelty, complexity, poor presentation and restricted setting, this draft cannot be accepted by ICLR.